# OpenReview forum: "Defending Against Unforeseen Failure Modes with Latent Adversarial Training"
_TMLR — Accepted by TMLR_

### Review · Reviewer_H6Wb · 2025-04-18

**Summary Of Contributions:**

### **Summary**
This paper introduces latent adversarial training to remove harmful behaviors from models. The authors evaluate the performance of three approaches (random latent perturbation (RLP), adversarial training (AT), latent adversarial training (LAT)) in mitigating harmful behaviors of models across three scenarios (image classification, text classification, and text generation). The authors conducted extensive experiments to illustrate the specific performance of LAT, to some extent demonstrating the effectiveness of this approach in defending against unforeseen attacks.

**Audience:**

Yes

**Claims And Evidence:**

Yes

**Requested Changes:**

- **(Most important)** Overclaim: The authors need to clarify the specific scenarios of unforeseen failure modes and extend the evaluation to various attack methods (such as jailbreak[1], llm dos attacks[2], etc.) to strongly demonstrate the effectiveness of the proposed LAT method.
- **(Important)** Evaluations: The authors should conduct ablation experiments on the proposed LAT to evaluate its generalizability.
- **(Important)** Comparisons: The authors should compare the proposed LAT method with corresponding defense methods under different attack scenarios, highlighting their differences and performance. Alternatively, directly articulating the specific advantages of the LAT method is also acceptable.
-  **(Trivial)** Typeset: The authors are encouraged to reconstruct the figures for a clearer presentation of the experimental results.

[1] Jailbreaking Leading Safety-Aligned LLMs with Simple Adaptive Attacks.

[2] Denial-of-service poisoning attacks against large language models

**Strengths And Weaknesses:**

### **Strengths**
- Theoretically, LAT exhibits characteristics of a generalized defense mechanism with inherent attack-agnostic properties. This theoretical foundation substantiates the methodological rigor employed in its evaluations.
- The evaluations seems to prove the improvements of model robustness.
- The paper is easy to understand.
### **Weaknesses**
- The attack method considered in this paper (described as an unforeseen attack) is very simple and straightforward. This direct poisoning attack method in the training data seems to have been widely studied and should not be regarded as an unforeseen attack. Evaluating in this manner seems unable to fairly measure the performance of LAT. Moreover, there are many defense schemes against such poisoning attacks (whether it is image/text classification or text generation), and many of these schemes do not require additional training.
- This paper lacks evaluation baselines. The attack success rate of the considered attack and the performance of standard training seem not to be reflected in the evaluation. Furthermore, the analysis of the conclusions only stays on the surface of the results, lacking an in-depth analysis of the reasons for the specific performance (e.g. why AT outperforms LAT on BeaverTails.).
- This paper lacks an explanation of the rationale behind the choice of hyperparameters related to LAT (such as the attacked hidden layer) and corresponding ablation studies. Additionally, the introduction of each experiment is somewhat brief, omitting detailed explanations.
- The results of the experimental evaluation are somewhat cluttered and difficult to read.

Although I personally acknowledge the novelty and contribution of this paper, the insufficient experimentation and exaggerated descriptions render it unsuitable for publication in its current version.

---

### Review · Reviewer_EsHp · 2025-04-26

**Summary Of Contributions:**

This paper introduces latent adversarial training (LAT) to improve model robustness against unforeseen failure modes, such as trojans, jailbreaks, and novel adversarial attacks. Unlike traditional methods that perturb the input space, LAT targets hidden layer activations to uncover vulnerabilities that might otherwise go unnoticed. Experiments in image classification, text classification, and text generation show that LAT outperforms conventional adversarial training in both robustness against novel attacks and the ability to maintain clean performance.

**Audience:**

Yes

**Broader Impact Concerns:**

The author has discussed the broader impacts of the paper.

**Claims And Evidence:**

Yes

**Requested Changes:**

1. Provide a more formal explanation or even a small theoretical analysis of why perturbations in latent space would generalize robustness across different failure modes, especially unforeseen ones. A sketch of assumptions and an argument would be valuable.
2. Although mentioned in the Limitations, incorporating experiments on larger state-of-the-art LLMs (e.g., LLaMA-3, Qwen-2.5) or, at the very least, discussing the expected scaling behavior would enhance the work and make it more compelling.
3. Include an analysis of the training cost for LAT compared to standard AT and RLP, and discuss its implications for practical deployment, especially in resource-constrained settings.

**Strengths And Weaknesses:**

- Strength：

1. The paper addresses an important challenge in ensuring AI system reliability by targeting unforeseen failure modes.
2. It presents a thorough empirical evaluation across diverse modalities, including vision, text classification, and text generation, using a range of architectures such as CNNs, transformers, and LLMs.
3. The use of Pareto frontiers to analyze the trade-offs between clean accuracy and adversarial robustness is thoughtful and strengthens the experimental rigor.

- Weakness:

1. The theoretical justification for why latent adversarial perturbations should improve generalization to unforeseen attack distributions remains largely intuitive and is not rigorously developed.
2. The evaluation is limited to relatively small and outdated models, such as LLaMA-2-7B, raising concerns about whether the observed benefits of LAT extend to larger, state-of-the-art models like LLaMA-3 and Qwen-2.5, which are more widely deployed.
3. The computational overhead introduced by LAT, particularly due to latent-space perturbation generation, is not analyzed; given the importance of training efficiency for large-scale systems, this omission weakens the practical relevance of the work.

---

### Review · Reviewer_FdnN · 2025-04-30

**Summary Of Contributions:**

This paper tackles the challenge of making network-based models more robust against adversarial attacks. Specifically, the paper empirically analyzes how vision, language understanding, and language generation tasks behave when subject to such attacks and proposes to use LAT to improve their robustness. Empirically, the paper investigates trojans and novel classes of attacks.

**Audience:**

No

**Claims And Evidence:**

Yes

**Requested Changes:**

If the goal was to make a benchmark paper, my opinion is that having a few other methods to improve the robustness would help and increase the soundness of this work. Otherwise, I cannot see where the contribution on the methodology stands.

**Strengths And Weaknesses:**

## Main strengths:

1. The paper tackles an important challenge, namely how to improve robustness of SOTA models for language, vision tasks.
2. The paper compares the AT and LAT techniques, which differ on where the perturbation at training time happens (i.e., training set or latent space).
3. The experiments are well documented and results are clearly stated and motivated.
4. The text has good quality, although being redundant in some parts (e.g., the difference between AT and LAT is repeated multiple times).

## Main weaknesses:
1. The novelty is quite limited: the paper makes essentially an experimental comparison between two approaches for improving the robustness of a model over a few different types of data. This is positions half way between a benchmark and a paper with methodological novelties.
2. The impact is probably limited, and the paper is more in a shape of a workshop paper. However, some insights are interesting and useful to the community.

---

### Review · Reviewer_51Bc · 2025-06-11

**Summary Of Contributions:**

This paper introduces Latent Adversarial Training (LAT) as a promising method to enhance the robustness of AI systems against unforeseen failure modes, such as novel adversarial attacks and backdoors, without requiring explicit examples of these failures during training. The core idea is to leverage the compressed nature of latent representations, ***hypothesizing that "many failures that are difficult to elicit from the input space may be easier to elicit from the latent space."*** The authors demonstrate LAT's effectiveness across diverse tasks, including image classification, text classification, and text generation, showing that LAT, particularly when applied to an "optimal layer," often achieves Pareto dominance over traditional input-space Adversarial Training (AT) and Random Latent Perturbations (RLP) in terms of both clean data performance and robustness. The authors also claim that they are the first to explore LAT's capability in Large Language Models (LLMs) for removing harmful behaviors and evaluating robustness against both backdoors and novel attacks.

**Audience:**

Yes

**Claims And Evidence:**

Yes

**Requested Changes:**

First, to strengthen the methodological foundation, please integrate a more comprehensive suite of defense baselines that uses the latent and feature spaces. This expanded scope may include, but not be limited to, manifold-based defense [1], representation space regularization [2], self-supervised adversarial training [3], randomized smoothing [4], representation noising [5] and synonym encoding [6]. I acknowledge that considering image classification, text classification, and text generation tasks introduces a highly intensive workload due to their distinct baseline algorithms in different fields. However, to rigorously validate your claims for publication, expanding the experimental evaluation is absolutely essential. The current experiments, while yielding interesting findings, are insufficient to establish the broad applicability of your interesting idea.

After incorporating these additional defense methods, don't forget to conduct a more thorough hyperparameter search for AT and other baselines, analogous to the "optimal layer" sweep previously performed for LAT, to ensure optimal performance and fair comparisons.

[1] Textual Manifold-based Defense Against Natural Language Adversarial Examples. EMNLP 2022.

[2] Spectral regularization for adversarially-robust representation learning. arXiv:2405.17181.

[3] Removing Adversarial Noise in Class Activation Feature Space. ICCV 2021.

[4] Randomized Smoothing with Masked Inference for Adversarially Robust Text Classifications. ACL 2023.

[5] Representation Noising: A Defence Mechanism Against Harmful Finetuning. NeurIPS 2024.

[6] Natural Language Adversarial Defense through Synonym Encoding. UAI 2021.

**Strengths And Weaknesses:**

### Strengths
- The motivation (hypothesis) that "many failures that are difficult to elicit from the input space may be easier to elicit from the latent space" is interesting.
- The idea is (partly) supported by experiments including image classification, text classification, and text generation tasks. I appreciate the authors for this.
- The experimental results regarding the Pareto frontier seems significant.
---

### Weaknesses
- **Missing baselines**: The paper's primary comparison is limited to input-space Adversarial Training (AT) and Random Latent Perturbations (RLP). While these are relevant, the field of representation-space defenses is much broader. To robustly claim the "superiority of adversarial training in the representation space," the paper lacks comparison with several established and emerging methods that also operate in or leverage the latent/feature space.
- **Insufficient analysis**: The paper evaluates LAT across image classification, text classification, and text generation, but the discussion of fundamental differences in representation learning and adversarial vulnerability across these modalities, especially for text generation, could be more detailed. In particular,
    - Discrete vs. continuous inputs: The inherent discrete nature of text inputs (tokens) versus continuous image pixels leads to different challenges for adversarial attacks and defenses. While the paper notes the non-differentiability of tokenization, a deeper analysis of how this fundamentally shapes the design and effectiveness of latent space defenses (e.g., starting at the embedding layer) compared to continuous inputs is needed.
    - Text generation: The observation that residual stream perturbations are more effective than Q/K/V perturbations in Transformers is significant, but the paper could delve deeper into why this is the case, linking it to the complex interplay of attention mechanisms, information flow, and the aggregation of knowledge within the residual stream. The observed dataset sensitivity for backdoor removal in text generation (LAT outperforming AT on Anthropic-HH but AT outperforming LAT on BeaverTails) also warrants a more thorough investigation into the underlying reasons.
- **Fairness of experimental comparison**: The paper emphasizes that LAT's superior performance is often achieved by selecting an "optimal layer" through a sweep across different layers. This systematic optimization effort for LAT is not explicitly mirrored for AT or other potential baselines. The absence of a similarly exhaustive hyperparameter search or exploration of advanced variants for AT (e.g., different attack algorithms beyond PGD, adaptive AT variants, or more varied perturbation budgets) creates an imbalance in the comparison. LAT is presented in its most optimized form, while other methods might be operating under sub-optimal configurations.

---

### Review · Reviewer_MotU · 2025-06-24

**Summary Of Contributions:**

This paper investigates latent adversarial training as a method to defend unforeseen adversarial examples.
Specifically, they propose perturbing the feature space (i.e., the intermediate layers of the neural network) instead of perturbing the input space as in standard adversarial training.

**Audience:**

Yes

**Claims And Evidence:**

Yes

**Requested Changes:**

1. Make it clear in the main paper which layer is perturbated in latent adversarial training along with the perturbation radius.
And discuss how the layer is chosen.

**Strengths And Weaknesses:**

1. The method is quite intuitive and simple.
The experiments demonstrate that the propose method does expand the pareto frontier.
Though, the improvements in some cases are somewhat marginal (e.g., Figure 2a and Figure 2b.2).

1. There has to be a trade-off regarding which intermediate layer to perturb.
On the one hand, standard adversarial training in the input layer presumably are not able to capture unforseen adversarial examples, because the \\(\ell_p\\) norm may fail to measure semantic similarity.
On the other hand, perturbations in the final layer directly might change the semantic information drastically, and thus it is unreasonble to require the model's prediction to stay unchanged.

1. One drawback is that it is probably harder to tune the hyperparameters of latent adversarial training.
The perturbation radius in standard adversarial training usually has a good geometric interpretation.
For instand, a radius of \\(8 / 255\\) is a good starting point for \\(\ell_\infty\\)-norm standard adversarial training, because \\(8 / 255 \approx 3.1\\%\\) of the pixel range is unlikely to change the semantic meaning of the input image.
However, the perturbation radius in the intermediate layers is hard to interpret.
Thus, it may require some tuning efforts to find a good perturbation radius that yields good clean/robust accuracy, which is a hidden cost to use this method in practice.
It is also unclear which intermediate layer to perturb, which again needs tuning.

---

### Review · Reviewer_kNbg · 2025-06-30

**Summary Of Contributions:**

The paper explores how neural networks can be more robust against novel adversarial attacks. Specifically, the author's show that
 latent adversarial training (LAT) is an effective defence against unforeseen failure modes in AI systems, outperforming input-space AT in the image classification, text classification, and text generation domains.

**Audience:**

Yes

**Claims And Evidence:**

No

**Requested Changes:**

- I have set claims and evidence to "no" due to the absence of evidence for the claim that LAT is more efficient than AT. Please either remove this claim or, preferably, provide supporting evidence.

- Though I’m reluctant to request additional experiment given the three diverse modalities and the considerable scale of the models and datasets, I believe that a paper whose main claim is that one existing method outperforms its peers should benchmark against a broader range of state-of-the-art defenses. If the claim is confined to demonstrating that LAT outperforms AT on novel attacks, this work might be better suited to a workshop or short-conference format.

**Strengths And Weaknesses:**

Strengths:
- Overall I do like the paper, its presented well, provides an interesting story, and also has rather deep evaluations of LAT specifically.

In summary:
- The paper is very well presented, reading it is easy and very clear.
- The evaluations are relatively complete, multiple modalities and several ablations give valuable insight. Though, more comparison methods could have been presented.
- The insights are interesting  and I think considering novel attacks is both important and practical.

Weaknesses:
- Since the premise of the paper is preparing a model for novel attacks ("unknown unknowns"), knowing which is the optimal layer to attack with LAT is difficult. The same is true for the perturbation magnitude hyper-parameter.
- The authors claim that LAT is computationally cheaper than AT, however I cannot see supporting evidence reported in the paper (e.g., wall-clock time, memory footprint)
- Since the manuscript has no methodological contributions, its positioning is somewhat awkward. It could be a benchmark paper, however it has too few comparison methods and certain SoTA models are missing (e.g. ViT/Dino backbones for classification). The presented insights are interesting but, given LAT is already used fairly often, their impact could be questioned.

---

### Decision · Action_Editor_y23i · 2025-07-02

**Recommendation:** Accept as is

**Audience:**

Yes

**Audience Explanation:**

everyone agrees the topic matters; one review calls it an “important challenge in ensuring ai system reliability,” and another says tackling novel attacks is “important and practical.” the method slots right into existing AT pipelines, so practitioners can actually use it, not just cite it. with every reviewer marking audience = yes, the interest is clear, so acceptance makes sense

**Claims And Evidence:**

Yes

**Claims Explanation:**

the experiments are good: multiple reviewers point out that the paper “demonstrate[s] improvements across different tasks,” and another notes the “deep evaluations of LAT.” A couple folks still wish for more baselines, but the core trade-off of higher clean accuracy and more robustness shows up in vision, text-class, and llm runs, which is plenty to back the main claim. given that, the evidence requirement is met